# Thin Film Composite Membranes Based on the Polymer of Intrinsic Microporosity PIM-EA(Me_2_)-TB Blended with Matrimid^®^5218

**DOI:** 10.3390/membranes12090881

**Published:** 2022-09-13

**Authors:** Mariagiulia Longo, Marcello Monteleone, Elisa Esposito, Alessio Fuoco, Elena Tocci, Maria-Chiara Ferrari, Bibiana Comesaña-Gándara, Richard Malpass-Evans, Neil B. McKeown, Johannes C. Jansen

**Affiliations:** 1Institute on Membrane Technology, National Research Council of Italy (CNR-ITM), Via P. Bucci, 17/C, 87036 Rende, Italy; 2School of Engineering, University of Edinburgh, Robert Stevenson Road, Edinburgh EH9 3FB, UK; 3EaStCHEM, School of Chemistry, University of Edinburgh, David Brewster Road, Edinburgh EH9 3FJ, UK

**Keywords:** thin film composite membranes, gas separation, polymer blend, polymer of intrinsic microporosity

## Abstract

In this work, thin film composite (TFC) membranes were fabricated with the selective layer based on a blend of polyimide Matrimid^®^5218 and polymer of intrinsic microporosity (PIM) composed of Tröger’s base, TB, and dimethylethanoanthracene units, PIM-EA(Me_2_)-TB. The TFCs were prepared with different ratios of the two polymers and the effect of the PIM content in the blend of the gas transport properties was studied for pure He, H_2_, O_2_, N_2_, CH_4_, and CO_2_ using the well-known time lag method. The prepared TFC membranes were further characterized by IR spectroscopy and scanning electron microscopy (SEM). The role of the support properties for the TFC membrane preparation was analysed for four different commercial porous supports (Nanostone Water PV 350, Vladipor Fluoroplast 50, Synder PAN 30 kDa, and Sulzer PAN UF). The Sulzer PAN UF support with a relatively small pore size favoured the formation of a defect-free dense layer. All the TFC membranes supported on Sulzer PAN UF presented a synergistic enhancement in CO_2_ permeance, and CO_2_/CH_4_ and CO_2_/N_2_ ideal selectivity. The permeance increased about two orders of magnitude with respect to neat Matrimid, up to ca. 100 GPU, the ideal CO_2_/CH_4_ selectivity increased from approximately 10 to 14, and the CO_2_/N_2_ selectivity from approximately 20 to 26 compared to the thick dense reference membrane of PIM-EA(Me_2_)-TB. The TFC membranes exhibited lower CO_2_ permeances than expected on the basis of their thickness—most likely due to enhanced aging of thin films and to the low surface porosity of the support membrane, but a higher selectivity for the gas pairs CO_2_/N_2_, CO_2_/CH_4_, O_2_/N_2_, and H_2_/N_2_.

## 1. Introduction

Membrane-based gas separation processes are a technology in continuous evolution and growth. Their economic and environmental benefits, compared to the traditional industrial gas separation processes [1], have garnered the attention of many researchers, leading to the development of new materials used in various separation and purification processes, such as post-combustion capture of CO_2_ [2,3], enrichment of O_2_ from the air [4], removal of N_2_ from the air [5,6], and CO_2_ from natural gas or biogas [7,8]. Membranes need to have high permeability and good selectivity to reach high process efficiencies. However, it is not easy to find materials that satisfy these conditions, and this is currently a challenge where many efforts converge. Polymers of intrinsic microporosity (PIMs), synthesized for the first time by Budd and McKeown [9], are among the most promising candidates for preparing the new generation of membranes since their rigidity and physical–chemical requirements lead to highly-permselective properties. The high rigidity and contortion of the polymer chains prevent their packing, inducing the formation of high free volume that characterizes these materials. Since the introduction of archetypal PIM-1, numerous new PIMs with a variety of different chemical structures have been synthesized, resulting in ever-better-performing materials. However, the complexity of the synthesis, the high costs, and the aging over time have prevented the application of these materials on a large scale for industrial gas separation applications. 

Merkel et al. [10] demonstrated that permeance (rather than selectivity) controls the economics of large-scale CO_2_ capture processes. Thus, to reduce the capital cost, membranes with higher permeances are needed. For instance, doubling CO_2_ permeance without selectivity loss would roughly cut the required membrane area in half, which corresponds to a significant fraction of the fixed capital cost of the plant. Most PIM membranes presented in the literature were prepared as self-standing dense structures with relatively low permeance and requiring large amounts of polymer and a high cost of production per unit area. Theoretically, decreasing the membrane thickness allows for higher permeances (permeance = permeability/thickness) but at the expense of its mechanical stability and integrity. A feasible solution to obtain very thin defect-free membranes is to produce a thin film composite (TFC) structure, consisting of a thin selective layer on a porous support. The gas separation performance is controlled by the thin selective layer, while the porous support provides mechanical strength. TFC membranes can provide separations with high fluxes combined with high mechanical resistance [11].

Polymeric composite membranes have been manufactured with a diverse range of high-free volume materials, including perfluoropolymers [12,13,14,15]. In the case of expensive high-performance polymers, the use of very thin layers considerably reduces the overall membrane costs due to the small amount of polymer required. The selection of appropriate support is a crucial step to obtain good membranes. A combination of high porosity and small pore size is necessary to maximize the gas permeation and, simultaneously, minimize the penetration of the polymer solution into the support during the coating process. Other parameters, such as polymer concentration and type of solvent, should also be carefully considered for obtaining defect-free membranes. Different methods have been used to prepare TFC membranes, such as dip-coating, spin-coating, roll-coating, and kiss-coating [16,17,18,19]. In some cases, pre-wetting with a non-solvent is used to prevent the penetration of the polymer solution into the support [16]. Recently, Bhavsar et al. [20] produced PIM-1-based TFC membranes for CO_2_ separation on polyacrylonitrile (PAN) supports, adding highly permeable nanoparticulate fillers of hyper-crosslinked polystyrene, HCP, and its carbonized form, C-HCP. A very high filler loading of up to 60% was successfully incorporated within the TFC membranes but not in the equivalent self-standing dense films. The PIM-1/C-HCP TFC membranes at 60% loadings exceed the 2008 upper-bound defining the trade-off between permeability and selectivity. Bernardo et al. [21] used a PIM composed of Tröger’s base and ethanoanthracene (PIM-EA(H_2_)-TB) as the selective layer for coating PAN-based hollow fibre and PAN and Fluoroplast F-42-based flat sheet commercial supports. The pristine PAN-support yielded a lower selectivity than the self-standing dense membrane prepared with PIM-EA(H_2_)-TB. Functionalization of the PAN hollow fibre support, through the hydrolysis of nitrile groups to the corresponding –COOH groups, enhanced the compatibility with the PIM coating layers and increased the selectivities to the level of the self-standing dense membranes. Borisov et al. [19] reported synergetic enhancement of CO_2_/N_2_ selectivity and permeability of PIM-1 thin film composite membranes. The TFC membrane consisting of PIM-1 as a selective layer on the cross-linked PTMSP as the gutter layer and commercial microfiltration membrane MFFK-1 as the support layer revealed an increased CO_2_/N_2_ selectivity (from 18 to 35–55). 

The challenge to improve the competitiveness of membranes and their uptake on an industrial scale relies on the fabrication of inexpensive polymer membranes having high selectivity and excellent permeability. Therefore, blending of PIMs with commercial materials can be a valid strategy to obtain cheaper membranes compared to neat PIM membranes, with a good trade-off between permeability and selectivity. Therefore, in this study, we chose PIM-EA(Me_2_)-TB [22] (Figure 1) for its high permeability and excellent performance in some separations [22] and the commercial polymer Matrimid^®^5218 for its selectivity. Matrimid^®^5218 has already been used in blends with other PIMs [23,24,25,26], including the closely related PIM-EA(H_2_)-TB, structurally similar to the PIM used in this study, showing good compatibility [27]. Commercially available porous membranes were used as supports for the TFC membranes, based on their expected resistance to chloroform and relatively small pore sizes in the nanofiltration or low ultrafiltration range, with nominal pore sizes from approximately 20 to 50 nm.

## 2. Materials and Methods

### 2.1. Materials

The PIM-EA(Me_2_)-TB was prepared, starting from the 9,10-dimethyl-9,10-dihydro-2,6(7)-diamino-9,10-ethanoanthracene and following a procedure reported previously [22]. Commercial polymer Matrimid^®^5218 (Figure 1) was supplied by Huntsman (Basel, Switzerland). The polyvinylidene fluoride PV 350 membrane was purchased from Nanostone Water. The poly(vinylidenefluoride-co-tetrafluorethylene) Fluoroplast 50 membrane, the polyacrylonitrile PAN 30 kDa membrane, and the polyacrylonitrile PAN UF membrane were kindly supplied as free samples by JSC-STC “Vladipor” (Rus)(Vladimir, Russia), Synder Filtration (Vacaville, CA, USA) and Sulzer (CH) (Winterthur, Switzerland), respectively. The membranes with wetting agents were successively washed with water, methanol, isopropanol, and hexane, and were air-dried before being used as support for the preparation of the composite membranes. Their characteristics are given in Table 1. Chloroform for the TFC membrane preparation was used without further purification. The two-component polydimethylsiloxane (PDMS) resin ELASTOSIL^®^ M 4601 was purchased from Wacker Chemie AG. The gases for the permeation tests (hydrogen, helium, nitrogen, oxygen, methane, and carbon dioxide, all with purity of 99.99+%) were purchased from SAPIO, Monza MB, Italy.

### 2.2. Preparation of Matrimid^®^5218/PIM-EA(Me_2_)-TB Blend Membranes

The Matrimid^®^5218 and PIM-EA(Me_2_)-TB polymers were dried at ambient pressure at 110 °C for 48 h. Then a 2 wt.% mother solution of each polymer was prepared by weighing 1 g of polymer and 49 g of chloroform in an Erlenmeyer flask. The solution was stirred magnetically overnight to yield a coloured but visibly clear, apparently homogeneous solution. Then, 6 mL of each pure solution was transferred into two separate vials, using a glass-fibre syringe filter with a nominal pore size of 3 μm to remove dust and undissolved gel particles (if present). Three additional solutions were made with 25%/75%, 50%/50%, and 75%/25% of PIM-EA(Me_2_)-TB and Matrimid^®^5218 by filtering 1.5, 3, and 4.5 mL of the PIM-EA(Me_2_)-TB solution and 4.5, 3, and 1.5 mL of the Matrimid^®^5218 solution, respectively, into three additional vials (Table 2). The polymer solutions were stirred magnetically for at least 10 min to become entirely homogeneous. 

Samples of the porous support membranes (ca. 5 cm × 5 cm) were coated manually with the polymer solution using a plastic pipette (Figure 2). The membrane was kept in a near-horizontal position, and it was gradually wetted with the polymer solution by manually dripping the solution in a zig-zag motion from the high to the low end of the membrane in less than 10 s. The membrane was then kept in a vertical position to let the excess solution flow away. It was then placed horizontally on a heated surface (50 °C) to allow the membrane to dry completely. Depending on the polymer solution and the membrane type, this procedure resulted in visibly uniform or non-uniform coverage of the support membrane surface. Some membranes became shiny, indicating a continuous dense coating, whereas others became matt or patched, with shiny and matt areas, suggesting the presence of a still porous top layer (See Appendix A).

The ELASTOSIL^®^ M 4601, consisting of a mix of pre-polymer A and crosslinker B in weight ratio 9:1, was dissolved in cyclohexane to obtain a solution composed of 80 wt.% cyclohexane, 18 wt.% of the base polymer (component A), and 2 wt.% of curing agents (component B). The solution was heated at 60 °C and stirred for 1 h. After cooling, it was further diluted with cyclohexane to obtain a final concentration of 10 wt.%. To heal occasional pinhole defects, the solution was pipetted manually onto those membranes with abnormally high permeability and low selectivity, while slightly tilting the membranes to allow the excess solution to flow away. The membranes were left to dry for several days at room temperature to allow the polymer’s crosslinking and the solvent’s complete evaporation.

### 2.3. Membranes Characterization

#### 2.3.1. Supports and TFC Characterization

Morphological characterization of the membranes was performed by scanning electron microscopy (SEM) on a Phenom Pro X desktop SEM, equipped with a backscattering detector (BSD-Phenom-World B.V., Eindhoven, the Netherlands). Before the SEM analysis, all samples were sputter-coated with a thin layer of gold using a Quorum Q150 RS sputter machine, Quorum Technologies Ltd, Ashford, Kent, England (1 min cycle) to minimize the charge and improve the image quality. Infrared spectroscopy (FT-IR) analyses were performed on a Spectrum Spotlight Chemical Imaging Instrument (PerkinElmer, Waltham, MA, USA) with a universal ATR sampling accessory. 

#### 2.3.2. Transport Properties

Single gas permeation tests were carried out using a fixed-volume pressure increase instrument (designed by HZG, constructed by Elektro & Elektronik Service Reuter, Geesthacht, Germany), as described elsewhere [28]. Tests were carried out at 25 °C, with a feed pressure of 1 bar. An effective area of 2.14  cm^2^ was used for all membranes unless the flux was too high, in which case an area of 0.65 cm^2^ was used, and the gases were tested in the order H_2_, He, O_2_, N_2_, CH_4_, and CO_2_. 

The permeability, *P*, was calculated from the slope of the pressure versus time curve in steady state condition from the following equation [29]:(1)$$p_{t}=p_{0}+{\left(\frac{dp}{dt}\right)}_{0}\cdot t+\frac{RT\cdot A}{V_{P}\cdot V_{m}}\cdot \frac{p_{f}\cdot P}{l}\cdot \left(t-\frac{l^{2}}{6D}\right)$$
in which *p_t_* and *p*_0_ are the permeate pressures at time *t* and the start, respectively. *(d_p_/d_t_)*_0_ is the baseline slope, *R* is the universal gas constant, *T* is the absolute temperature, *A* is the membrane area, *V_P_* is the permeate volume, *V_m_* is the molar volume of the permeating gas at standard temperature and pressure (0 °C and 1 atm), *p_f_* is the feed pressure, and *l* is the membrane thickness (considering the thickness of 1 µm). Permeabilities (*P*) are reported in Barrer (1 Barrer = 10^−10^ cm^3^ (_STP_) cm cm^−2^ s^−1^ cmHg^−1^). 

The permeance, *Π*, was calculated from the slope of the pressure versus time curve in the steady state condition and reported in GPU (1 GPU = 10^−6^ cm^3^ _STP_ cm^−2^ s^−1^ cmHg^−1^). The ideal selectivity is the ratio of permeance of two species:(2)$$\alpha_{\left(A/B\right)}=\Pi_{A}/\Pi_{B}$$

## 3. Results and Discussion

### 3.1. Porous Support Characterization

The characteristics of the porous supports used to prepare TFC membranes are given in Table 1. The Nanostone Water PV 350, Vladipor Fluoroplast 50, Synder PAN 30 kDa, and Sulzer PAN UF present an average pore size in the nanofiltration or lower ultrafiltration range with values of 50, 50, 30, and 20 nm, respectively. The SEM analysis of the upper surfaces for all porous supports was carried out to better understand the suitability of the support for the TFC membrane preparation (Figure 3).

The surface of the Vladipor Fluoroplast shows a narrow sponge-like surface structure. The other supports present smooth surfaces.

### 3.2. TFC Membranes Characterization

Figure 4 shows the cross-sectional SEM images of the two series of selective TFC membranes on the Sulzer PAN UF and the Synder PAN 30 kDa porous supports with the neat Matrimid^®^5218, neat PIM-EA(Me_2_)-TB and their blends (see Appendix A for the other series). From the SEM image, the selective polymer layers seem to have good adhesion on the Synder PAN supports but visibly detach from the support in the case of pure Matrimid^®^5218, and Matrimid^®^5218 with 25% and 50% of the PIM on the Sulzer PAN UF supports. Although this may have occurred also as a result of the mechanical stresses on the films during sample preparation for the SEM analysis.

The presence of the neat PIM-EA(Me_2_)-TB coating layer on the porous PAN support was confirmed by the FTIR analysis of the composite membrane (Figure 5).

After coating, the characteristic peak at 2243 cm^−1^ due to the nitrile stretching the vibration of poly-acrylonitrile [30] is absent, confirming an adequate deposition of the neat PIM-EA(Me_2_)-TB layer since the –CN group is not present in its chemical structure. The most characteristic peaks of PIM-EA(Me_2_)-TB are the CH_2_ and CH_3_ asymmetric stretch vibrations of the methanoanthracene (EA) unit at 2960 cm^−1^ and the scissoring vibrations at 1420 cm^−1^ [27]. The broad water absorption region (around 3370 cm^−1^) in the neat PIM-EA(Me_2_)-TB demonstrates the relatively hygroscopic nature of TB-PIMs [22]. The thickness observed for the selective layer of the blends on the Sulzer PAN is around 4.5 µm, whereas it is around 2 µm for pure polymers. Based on the SEM observations, the thickness of the selective layer deposited on the Synder PAN 30 kDa membrane is about 4 µm for each TFC membrane except for the blend with the 75 wt.% of PIM ratio, for which the thickness measured is around 8 µm.

### 3.3. Pure Gas Transport Properties

The diagrams in Figure 6 summarize the gas transport properties for all TFC membranes prepared with the neat polymers and their respective blends on the four different supports. The selectivity for the gas pairs CO_2_/CH_4_, CO_2_/N_2_, O_2_/N_2_, and H_2_/N_2_ are reported as a function of the permeances of the most permeable species in the gas pair. The membranes prepared on the Sulzer PAN are the best performing TFCs (blue symbols in Figure 6 and Appendix A) with a synergic enhancement in the permeances for all gases in the blends compared to that of neat Matrimid, and a higher CO_2_/CH_4_ and CO_2_/N_2_ selectivity compared to the neat PIM-EA(Me_2_)-TB. Increasing the PIM-EA(Me_2_)-TB concentration in the blend, the CO_2_ permeance increases from 3.37 GPU to 129 GPU, the O_2_ permeance from 0.69 GPU to 33 GPU, and the H_2_ permeance from 7.08 GPU to 403 GPU. These values are up to two orders of magnitude higher than those of the neat Matrimid membranes (1.64 GPU for CO_2_, 0.33 GPU for O_2_, and 4.81 GPU for H_2_, respectively). This result demonstrates that the PIM-EA(Me_2_)-TB offers the possibility to tailor the permeability of Matrimid^®^5218 over a wide range. The gas transport properties for the neat TFC membranes prepared on Sulzer PAN present higher CO_2_ permeance and higher selectivity (*Π*_CO2_ ~ 130 GPU; CO_2_/N_2_ ~ 26) compared to that the TFC membranes prepared with the closely structurally related PIM-EA(H_2_)-TB (*Π*_CO2_ ~ 77 GPU; CO_2_/N_2_ ~ 10), reported in the literature [27]. Further, the selectivities for all gas pairs, CO_2_/CH_4_, CO_2_/N_2,_ O_2_/N_2_, and H_2_/N_2_, are higher than the corresponding values of the thick dense neat PIM-EA(Me_2_)-TB reference membrane (yellow symbols in Figure 6, methanol-treated by Carta et al. [22] and normalized to a thickness of 1 µm for easier comparison). The higher selectivity suggests faster physical aging of the polymers in thin films, as previously described [29], but could also be due to the dense structure of the TFCs, which are not subjected to the methanol treatment. 

In contrast to the PAN-based TFCs, the membranes prepared with the PVDF and Fluoroplast supports all have low or no selectivity (Figure 6, Appendix A). This may be due to their larger nominal pore size of about 50 nm. Probably, the pore size of 50 nm causes a non-uniform coating due to the partial infiltration of the polymer solution into the support. SEM images of the top surfaces confirmed an inhomogeneous coating with regions containing visible pores (See Appendix A) and, in other cases, the formation of evident cracks.

The TFCs based on the Fluoroplast and the PVDF supports have an H_2_/N_2_ selectivity of around 4 for most compositions of the coating solution. This value corresponds to the ratio of the square root of their molar masses, √(28/4), typical for Knudsen diffusion through small pores rather than selective transport through a dense film. It confirms the porous and defective nature of the membranes since the with Matrimid, and Knudsen diffusion is the dominant transport mechanism. Apparently, this affects mostly the least permeable gases because the CO_2_ and O_2_ permeability remain below 1000 GPU. This suggests that most of the support pores are correctly coated by the PIM (or its blends with Matrimid) and that only a relatively small fraction of nonselective pores remains. Further studies on these materials are needed to understand whether they can be successfully coated by slight changes in the membrane preparation protocol.

## 4. Conclusions

The present work demonstrates the successful preparation of thin-film composite membranes of PIM-EA(Me_2_)-TB and its blend with Matrimid^®^5218 on the porous PAN support membranes. The TFC membranes have higher selectivity than PIM-EA(Me_2_)-TB but also have much greater permeance than pure Matrimid^®^5218. The CO_2_/CH_4_ and CO_2_/N_2_ selectivity are doubled compared to the neat PIM-EA(Me_2_)-TB, reaching reasonable CO_2_ permeance of around 100 GPU. The higher selectivities of the TFC, compared to the corresponding values for the thick film of PIM-EA(Me_2_)-TB, are likely due to the accelerated aging of the thin PIM film on the TFC, and/or to the presence of residual casting solvent and consequently the more compact film, which was not methanol treated.

Supports of PVDF and Fluoroplast with a nominal pore size of about 50 nm yield poorly selective films, most likely due to the ineffective coating of the largest pores or to the poor quality of the thin dense polymer film, subject to crack formation. Blending with PIM-EA(Me_2_)-TB offers the possibility to tailor the permeability of Matrimid^®^5218 over a wide range and opens perspectives for making high permeability thin film composite membranes with the mechanical resistance of Matrimid^®^5218 and PIM-like permeabilities.

This study highlights the importance of the right choice of porous supports to guarantee the effective coating with a thin selective dense layer and for the successful preparation of defect-free TFC membranes with a low resistance to gas transport. Additional work is needed to further enhance the permeance of the membranes, maintaining high selectivity.

## Figures and Tables

**Figure 1 membranes-12-00881-f001:**
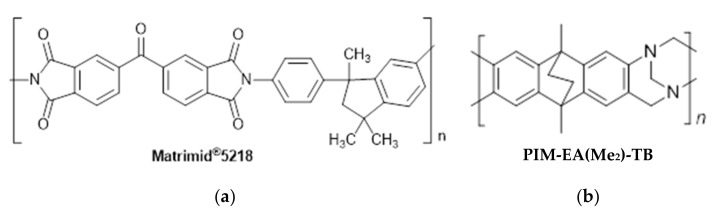
(**a**) Chemical structures of Matrimid^®^5218 and (**b**) PIM-EA(Me_2_)-TB.

**Figure 2 membranes-12-00881-f002:**
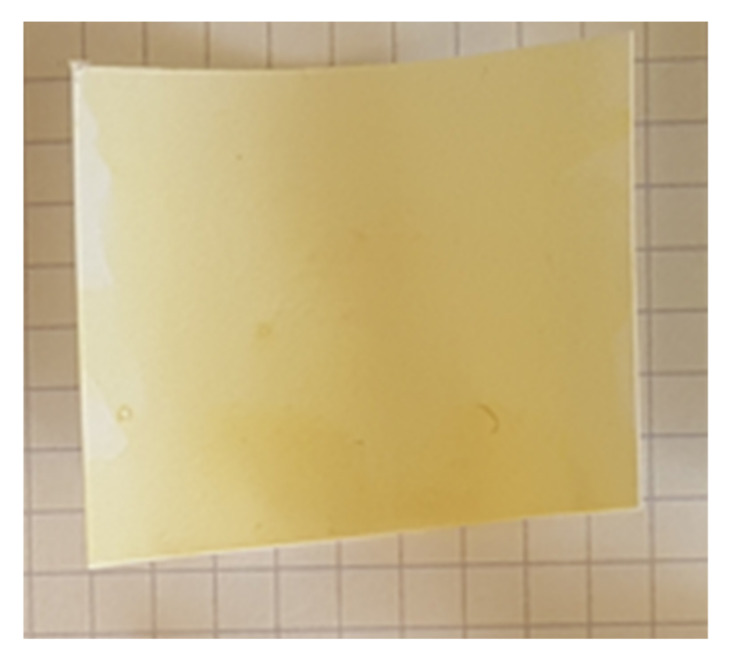
Example of a TFC membrane prepared by coating the Sulzer-PAN support (ca. 5 cm × 5 cm) with a blend of PIM-EA(Me_2_)-TB and Matrimid^®^5218 at 2 wt.% polymer solution.

**Figure 3 membranes-12-00881-f003:**

SEM images of the upper surface for the porous membranes used as supports for the TFC. preparation. The imagines were acquired after Au metallization with a magnification of 10,000× and a primary beam voltage of 10 KV. The scale bar is identical for all samples.

**Figure 4 membranes-12-00881-f004:**
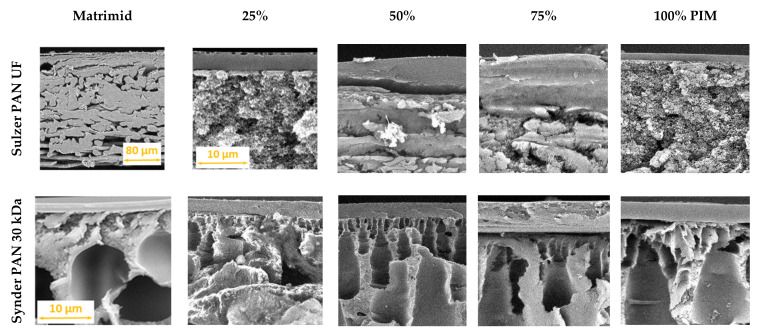
Cross-sectional SEM images of the Matrimid^®^5218/PIM-EA(Me_2_)-TB blend membranes with different blend compositions on a Sulzer PAN UF membrane support (top) and a Synder PAN 30 kDa porous membrane support (bottom). Magnification 5000× and electron acceleration voltage 10 kV. The scale bar is 10 µm for all samples, with the exception of the TFC of Matrimid on Sulzer PAN.

**Figure 5 membranes-12-00881-f005:**
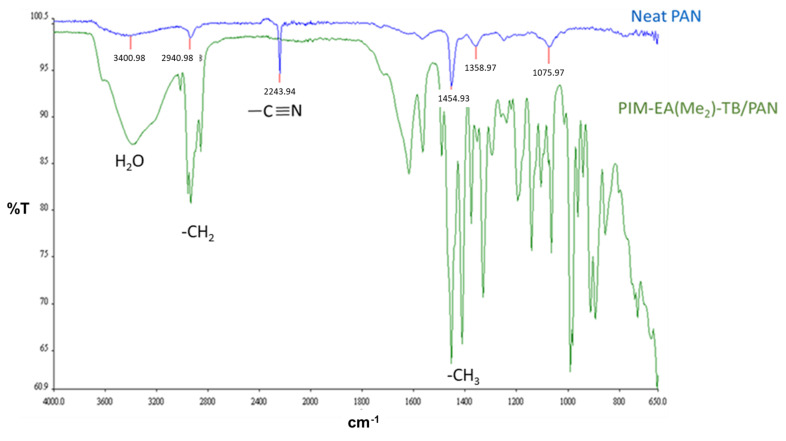
Attenuated total reflection FT-IR spectra of the top surface of the pristine Synder PAN 30 kDa porous support (blue), and the support with the neat PIM-EA(Me_2_)-TB coating (green).

**Figure 6 membranes-12-00881-f006:**
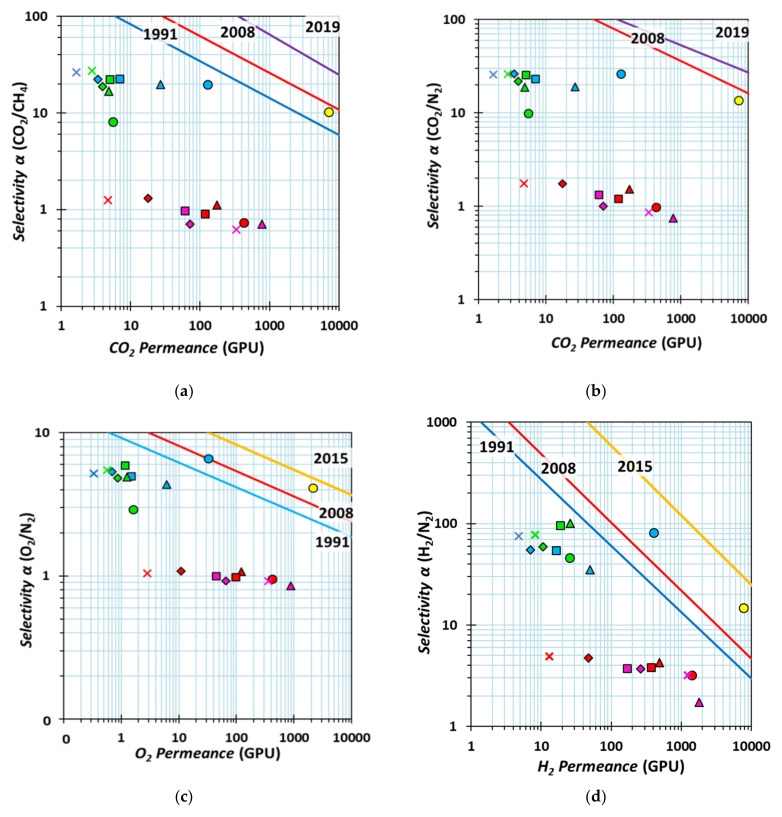
Selectivity plotted as a function of permeance for Matrimid^®^5218/PIM-EA(Me_2_)-TB blend membranes with different blend compositions for a number of relevant gas pairs: CO_2_/CH_4_ (**a**), CO_2_/N_2_ (**b**), O_2_/N_2_ (**c**), H_2_/N_2_ (**d**). Different colours are used for different supports: magenta: Nanostone Water PVDF PV350; red: Vladipor Fluoroplast 50 nm membrane; blue: Sulzer PAN UF membrane; green: Synder PAN 30 kDa membrane; Matrimid/PIM ratios: 00:100 (⬤), 25:75 (▲), 50:50 (■), 75:25 (◆), 100:00 (×). The reference data for a MeOH treated dense film of PIM-EA(Me_2_)-TB [22], normalized to a thickness of 1 µm are indicated for comparison (yellow circles, 🟡). The lines represent the maximum performance, based on the reported upper bounds: purple line (2019), yellow line (2015), red lines (2008), and blue lines for the previously proposed (1991) upper bounds [30,31,32], assuming an effective membrane thickness of 1 µm.

**Table 1 membranes-12-00881-t001:** Porous support membranes used for the TFC preparation.

Support Type Polymer	Polymer	Nominal (nm)
Nanostone Water, PV 350	Polyvinylidene fluoride	50
Vladipor Fluoroplast 50	Poly(vinylidenefluoride-co-tetrafluorethylene)	50
Synder PAN 30 kDa	Polyacrylonitrile	30
Sulzer PAN UF	Polyacrylonitrile	20

**Table 2 membranes-12-00881-t002:** Solutions of PIM-EA-(Me2)-TB and Matrimid^®^5218.

Matrimid/PIM Ratios (wt%)	PIM-EA(Me_2_)-TB Solution (mL)	Matrimid Solution (mL)
0/100	0	6.0
25/75	1.5	4.5
50/50	3.0	3.0
75/25	4.5	1.5
100/0	6.0	0

## Data Availability

Original data are available from the authors upon request.

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
