# Peer review of "Thin Film Composite Membranes Based on the Polymer of Intrinsic Microporosity PIM-EA(Me2)-TB Blended with Matrimid®5218"

_membranes, 2022, doi:10.3390/membranes12090881_

Round 1

Reviewer 1 Report

The author did a very interesting work that thin film composite (TFC) membranes, prepared by a blend of polyimide Matrimid®5218 and polymer of intrinsic microporosity (PIM), were used for gas separation. There are some comments that need to be considered by the authors. 

1.     The authors qualitatively describe by cross-sectional SEM that the adhesion between the dense layer and different supports is different. The authors should quantitatively demonstrate the difference in adhesion by other means of characterization.

2.     The authors show that the selected supports are of two types: nanofiltration and ultrafiltration. However, it can be seen from the molecular weight cut-off and surface SEM images that there is no nanofiltration support.

3.     The authors should compare the membrane performance in this study with the upper-bond of gas separation membranes and other up-to-date report like DOI: 10.1073/pnas.2114964119 ; 10.1016/j.eng.2022.03.016 for clarifying the fabricated membrane performance.

4.     Membrane characterization means are too few. The authors need to supplement a large number of characterization methods to demonstrate the structure and physicochemical properties of the membrane.

Author Response

We would like to thank the reviewer and we reported our answers to the questions below (further, we attached the review report in word format) :

  1. The authors qualitatively describe by cross-sectional SEM that the adhesion between the dense layer and different supports is different. The authors should quantitatively demonstrate the difference in adhesion by other means of characterization.

It must be said that the sample preparation method subjects the TFC to strong local stresses, and therefore, part of the skin detachment observed by SEM analysis may have been induced by the sample preparation method itself. Therefore, it would indeed be interesting to have a quantitative idea about the adhesion of the coated film to the porous substrate. However, we are not sure which kind of characterization method the reviewer has in mind, and to the best of our knowledge, the only quantitative analysis of the adhesion between two layers would be a peeling test. We strongly doubt whether this is feasible with a few micrometres thick film, since it would not be possible to detach a large enough piece of coating from the surface to fix it into the grips of the peeling test machine. Alternatively, perhaps it could be feasible to do a peeling test after first sticking a piece of pressure-sensitive adhesive tape on the surface and peeling this off, along with the thin film. We are not aware if this has been done before with membranes and in any case, also with this method we doubt how quantitative this would be, since it would certainly be affected by the properties of the much thicker adhesive tape itself. Therefore, we do not think that the request of the reviewer is realistic for these specific samples. The sentence relating to the “adhesion” from 242-246 line was changed in the following way and highlighted in yellow: “From the SEM image, the selective polymer layers seem to have good adhesion on the Synder PAN supports but visibly detach from the support in the case of pure Matrimid® 5218, and Matrimid® 5218 with 25% and 50% of the PIM on the Sulzer PAN UF supports. Although this may have occurred also as a result of the mechanical stresses on the films during sample preparation for the SEM analysis”

  1. The authors show that the selected supports are of two types: nanofiltration and ultrafiltration. However, it can be seen from the molecular weight cut-off and surface SEM images that there is no nanofiltration support.

The authors refer to the classification given to the membranes by the producers. The reviewer might have misunderstood the data in Table 1. These are not the molecular weight cut-off values (usually in kDa), but the producer-specified nominal pore sizes (in nm). In any case, the Vladipor membrane has clearly large pores in the UF or even MF range, but for the other membranes, the resolution of the Phenom Desktop SEM is insufficient to see the actual pores. Since the SEM is equipped with backscattering detector, it may observe also larger features that are located slightly below the surface of the selective skin layer.

  1. The authors should compare the membrane performance in this study with the upper-bond of gas separation membranes and other up-to-date report like DOI: 10.1073/pnas.2114964119 ; 10.1016/j.eng.2022.03.016 for clarifying the fabricated membrane performance.

We thank the reviewer for this suggestion. Strictly, it is not possible to compare our performances directly with the upper bounds, because these are based on the selectivity of a gas pair as a function of the permeability coefficient (a material property, in Barrer) of the usually most permeable gas, whereas we report the permeance (a membrane property, in GPU), which is thickness-dependent. Therefore, we plotted a simulated upper bound for membranes with a hypothetical thickness of 1 micron (Fig.6 in the main text).

(a)

(b)

(c)

(d)

Figure 6. Selectivity plotted as a function of permeance for Matrimid® 5218 / PIM-EA(Me2)-TB blend membranes with different blend compositions for a number of relevant gas pairs. Different colours are used for different supports: magenta: Nanostone Water PVDF PV350; red: Vladipor Fluoroplast 50 nm membrane; blue: Sulzer PAN UF membrane; green: Synder PAN 30kDa membrane; Matrimid/PIM ratios: 00:100 (˜), 25:75 (â–²), 50:50 (¢), 75:25 (¿), 100:00 (×). The reference data for a MeOH treated dense film of PIM-EA(Me2)-TB [22], normalized to a thickness of 1 µm are indicated for comparison (yellow circles, ˜). The lines represent the maximum performance, based on the reported upper bounds : purple line (2019), yellow line (2015), red lines (2008), and blue lines for the previously proposed (1991) upper bounds. [30–32], assuming an effective membrane thickness of 1 µm.

Moreover, we thank the reviewer for the suggested papers, but both DOI: 10.1073/pnas.2114964119; 10.1016/j.eng.2022.03.016 refer to mixed matrix membranes for gas separation and not to thin film composite membranes reporting the 2008 upper bound. Unfortunately, this is outdated and we prefer to report even the 2019 and 2015 upper bounds.

  1. Membrane characterization means are too few. The authors need to supplement a large number of characterization methods to demonstrate the structure and physicochemical properties of the membrane.

We thank the reviewer for the suggestion, but we think this is neither relevant, nor feasible within the time given for the revision of the manuscript. SEM analysis is the best possible structural/morphological analysis, as it directly observes the coated film and the gas permeation measurements are the single most relevant technique for the analysis of the membrane performance. These two techniques are further supported by the ATR-FT-IR analysis to confirm the presence and the composition of the coating layer. As further analysis, we could think of contact angle measurements but this is irrelevant or only marginally relevant for the gas transport properties. Contact angle measurements of the porous substrates could give some indication about their hydrophobicity, but it is well known that this analysis is also strongly affected by the surface roughness. Therefore, it provides only qualitative information and it is questionable whether this can be related to the membrane performance.

Reviewer 2 Report

The current manuscript describes the “Thin film composite membranes based on the polymer of in- 2 trinsic microporosity PIM-EA(Me2)-TB blended with Matri- 3 mid®5218”. In this study, thin film composite (TFC) membranes were fabricated with the selective layer based on a blend of polyimide Matrimid®5218 and polymer of intrinsic microporosity (PIM) composed of Tröger's base, TB, and dimethylethanoanthracene units, PIM-EA(Me2)-TB. The TFCs were prepared with different ratios of the two polymers and the effect of the PIM content in the blend on the gas transport properties was studied for pure He, H2, O2, N2, CH4, and CO2 using the lag method. After reviewing the manuscript reviewer pointed out that there is a lot of scope for improvement of this manuscript. A lot of sections may need further illustration; and more importantly, because this type of work has been already reported by a lot of researchers using different types of materials. One important point is that the abstract need to be revised completely.

The authors should consider critically these comments to improve the quality of the work. 

Few examples of where authors should make changes are provided below:

  1. The abstract is not showing the exact findings and output. Authors have to rewrite the abstract.
  2. In the introduction para two have to be revised with relevant literature related to current paper-like, Chemical Engineering Research & Design, 102 (2015) 297-306.; J. Cleaner Production, 133 (2016) 1008-1016.; Journal of Environmental Chemical Engineering 9 (2021) 104774.; Separation and Purification Technology, 211 (2019) 401–407.; Chem. Eng. J., 334 (2018) 2450–2458.;
  3.  3th para needs to rewrite again, Authors have to give the proper citation for sentence one like J. Mater. Chem. A, 6 (2018) 24569-24579.; Microchemical Journal 132 (2017) 36-42.; Chin. J. Chem. Eng. 25 (2017) 278–287 etc.
  4. All material details have to be added in Section 2.1 Materials.
  5. Explain the Figure 2 explanation and add all membrane images.
  6. All equations need to check and need the equation number in the manuscript.
  7. In figure 3, the high-resolution scale should be there on the SEM images.
  8. Figure 4; there is no scale on all images. It's very difficult to understand the image results.
  9. Figure 5; after coating the peak have to be labelled and needs to justify properly in the manuscript.
  10. If possible justify the pressure effect on the membrane performance.
  11. Grammar needs to check one more time.

Author Response

We would like to thank the reviewer and we reported our answers to the questions below (further, we attached the review report in word format): 

The current manuscript describes the “Thin film composite membranes based on the polymer of in- 2 trinsic microporosity PIM-EA(Me2)-TB blended with Matri- 3 mid®5218”. In this study, thin film composite (TFC) membranes were fabricated with the selective layer based on a blend of polyimide Matrimid®5218 and polymer of intrinsic microporosity (PIM) composed of Tröger's base, TB, and dimethylethanoanthracene units, PIM-EA(Me2)-TB. The TFCs were prepared with different ratios of the two polymers and the effect of the PIM content in the blend on the gas transport properties was studied for pure He, H2, O2, N2, CH4, and CO2 using the lag method. After reviewing the manuscript reviewer pointed out that there is a lot of scope for improvement of this manuscript. A lot of sections may need further illustration; and more importantly, because this type of work has been already reported by a lot of researchers using different types of materials. One important point is that the abstract need to be revised completely.

The authors should consider critically these comments to improve the quality of the work. 

Few examples of where authors should make changes are provided below:

  1. The abstract is not showing the exact findings and output. Authors have to rewrite the abstract.

The abstract already contains very detailed information, such as “The permeance increased about two orders of magnitude with respect to neat Matrimid, up to ca. 100 GPU, and the ideal CO2/CH4 selectivity increased from approximately 10 to 14 and the CO2/N2 selectivity from approximately 20 to 26 compared to the thick, dense reference membrane of PIM-EA(Me2)-TB“, and some more general evaluations. We do not really understand what other information the reviewer would like to see in the abstract.  

  1. In the introduction para two have to be revised with relevant literature related to current paper-like, Chemical Engineering Research & Design, 102 (2015) 297-306.; J. Cleaner Production, 133 (2016) 1008-1016.; Journal of Environmental Chemical Engineering 9 (2021) 104774.; Separation and Purification Technology, 211 (2019) 401–407.; Chem. Eng. J., 334 (2018) 2450–2458.;

The second paragraph of the introduction briefly introduces high free volume polymers, but is then specifically focussed on TFCs of PIM-based systems (pure PIMs, PIM-based MMMs). All references suggested by the reviewer are much broader (including even interfacial polymerization to obtain polyamides or membranes for water vapour removal from gas streams) and do not add information that is directly relevant for our manuscript. We have only checked if any new manuscripts have been published since our original submission, but we do not see any reason to revise the introduction at this point.

  1. 3th para needs to rewrite again, Authors have to give the proper citation for sentence one like J. Mater. Chem. A, 6 (2018) 24569-24579.; Microchemical Journal 132 (2017) 36-42.; Chin. J. Chem. Eng. 25

See also point 2. Again the reviewer asks for citations to irrelevant work on water vapour transport with MMMs based on nanoparticles, on facilitated transport or on membranes prepared by interfacial polymerization. Also in this case the authors of the works are Pravin G. Ingole and Hyung-Keun Lee and others.

  1. All material details have to be added in Section 2.1 Materials.

In section 2.1 all details about polymers and porous support used were revised and the details about the gas used for gas transport determination were reported in the following way: “(Hydrogen, helium, nitrogen, oxygen, methane and carbon dioxide, all with purity of 99.99+ %) were purchased from Sapio (Italy).”

  1. Explain the Figure 2 explanation and add all membrane images.

In fig.2, we reported a representative membrane. In the SI Figure 1. We have reported all membrane images of the TFC membranes with the four different supports after the permeability tests. In fact, you can see the O-ring of permeability instrument.

  1. All equations need to check and need the equation number in the manuscript.

Some numbers were not visible due to the automatic formatting by the MDPI template. We apologize that we have not seen this before. All equations are now numbered and the number is cited in the text where needed. The number is not cited in those cases where the equation follows immediately the reference, for instance ”…..was determined with the time-lag method according to the equation:”

  1. In figure 3, the high-resolution scale should be there on the SEM images.

We improved the visibility of the scale bar. Unfortunately, the text was not visible in the original submission due to the automatic formatting of the text box by the Membranes document template. We apologize for this.

  1. Figure 4; there is no scale on all images. It's very difficult to understand the image results.

The SEM imagines scale are reported in the Figure 4 and the visibility was improved. For all the images the scale bar is the same (10 µm) with the exception of the TFC of Matrimid on Sulzer PAN. This is specify now in the figure caption. 

  1. Figure 5; after coating the peak have to be labelled and needs to justify properly in the manuscript.

Thanks to the referee for his suggestion, we improved the fig.5 in the following way and the new figure was reported in the main manuscript.  

Moreover, the following sentence was revised and reported in yellow in the main manuscript: “After coating, the characteristic peak at 2243 cm-1 due to the nitrile stretching vibration of poly-acrylonitrile [30] is absent confirming an adequate deposition of the neat PIM-EA(Me2)-TB layer, since the –CN group is not present in its chemical structure”

If possible justify the pressure effect on the membrane performance.

We are not sure about the request of the reviewer because we carried out the gas transport measurements only at a constant pressure. If this is a suggestion for future work, we thank the reviewer for the suggestion. Indeed, after further optimization of the membranes, our future studies will be focused on the effect of the pressure on the membrane performance.

  1. Grammar needs to check one more time.

The grammar was checked in all the manuscript.

Round 2

Reviewer 1 Report

I have no more questions

Reviewer 2 Report

The comment number 10 has to be considered.